# Antioxidant Molecular Brain Changes Parallel Adaptive Cardiovascular Response to Forced Running in Mice

**DOI:** 10.3390/antiox11101891

**Published:** 2022-09-23

**Authors:** Clara Bartra, Lars Andre Jager, Anna Alcarraz, Aline Meza-Ramos, Gemma Sangüesa, Rubén Corpas, Eduard Guasch, Montserrat Batlle, Coral Sanfeliu

**Affiliations:** 1Institut d’Investigacions Biomèdiques de Barcelona (IIBB), Consejo Superior de Investigaciones Científicas (CSIC), 08036 Barcelona, Spain; 2Institut d’Investigacions Biomèdiques August Pi i Sunyer (IDIBAPS), 08036 Barcelona, Spain; 3Arrhythmia Unit, Hospital Clínic de Barcelona, 08036 Barcelona, Spain; 4Centro de Investigación Biomédica en Red-Cardiovascular (CIBERCV), 28029 Madrid, Spain

**Keywords:** physical exercise, proteasome, catalase, redox homeostasis, cardiac remodeling, brain resilience, young male mice, treadmill running

## Abstract

Physically active lifestyle has huge implications for the health and well-being of people of all ages. However, excessive training can lead to severe cardiovascular events such as heart fibrosis and arrhythmia. In addition, strenuous exercise may impair brain plasticity. Here we investigate the presence of any deleterious effects induced by chronic high-intensity exercise, although not reaching exhaustion. We analyzed cardiovascular, cognitive, and cerebral molecular changes in young adult male mice submitted to treadmill running for eight weeks at moderate or high-intensity regimens compared to sedentary mice. Exercised mice showed decreased weight gain, which was significant for the high-intensity group. Exercised mice showed cardiac hypertrophy but with no signs of hemodynamic overload. No morphological changes in the descending aorta were observed, either. High-intensity training induced a decrease in heart rate and an increase in motor skills. However, it did not impair recognition or spatial memory, and, accordingly, the expression of hippocampal and cerebral cortical neuroplasticity markers was maintained. Interestingly, proteasome enzymatic activity increased in the cerebral cortex of all trained mice, and catalase expression was significantly increased in the high-intensity group; both first-line mechanisms contribute to maintaining redox homeostasis. Therefore, physical exercise at an intensity that induces adaptive cardiovascular changes parallels increases in antioxidant defenses to prevent brain damage.

## 1. Introduction

A physically active lifestyle as contraposed to sedentary habits has huge implications for the health and well-being of individuals at all ages. Minimum weekly amounts of physical exercise recommended for each age (i.e., Physical Activity Guidelines for Americans [1]) can facilitate neural development at young age and adolescence and maintain mental health and cognitive capacities at middle and old ages [2,3,4,5]. Unfortunately, the current trend in our society is the increase in sedentary behavior [6,7]. Beneficial molecular changes induced by exercise in the brain include the activation of neurotrophic, antioxidant and anti-inflammatory pathways, improved proteostasis and epigenetic modifications [8,9,10,11,12]. Furthermore, exercise-induced cardiovascular and metabolic fitness contributes to brain health and perception of well-being [13,14]. Physical exercise, with its multiple systemic and cerebral effects and benefits, has been shown to be a non-pharmacological preventive therapy against cognitive decline and a range of chronic illnesses associated with advancing age [15,16,17]. In contrast, inactive adults aged in the ranges of 40 to 69 and 70 to older had shown an increased risk of premature death compared to those sufficiently active, according to a National Health survey in the United States [18]. At the other end of the indisputable health benefits induced by physical activity, there is a concern on harmful cardiovascular effects caused by excessive training. For instance, elite athletes or those highly motivated for sustained and intense workouts may suffer from atrial and ventricular arrhythmia owing to myocardial fibrosis [19,20]. However, severe cardiovascular events and sudden cardiac death cases in otherwise healthy subjects are rare [21]. Currently, findings of lower cognitive performance late in life after long-term strenuous exercise are scarce and require further study [22]. In addition, intense training may not increase cognitive benefices over moderate training according to experimental studies [23]. Therefore, there is a consensus about the body and brain benefices of chronic moderate physical exercise but there is uncertainty about the threshold of pernicious excessive training.

Redox mechanisms are proposed as main drivers of adaptive mechanisms and responses to physical exercise [24]. Low level of reactive oxygen species (ROS) generated by chronic aerobic or endurance training would elicit a down-stream signaling cascade leading to upregulation of antioxidant defense genes and a range of beneficial physiological effects [25,26,27]. However, excessive training or a pathological condition disrupting redox homeostasis [28,29] would generate high ROS levels leading to harmful oxidative reactions in proteins, lipids and nucleic acids. Similar exercise-induced redox mechanisms and adaptive responses have been reported in skeletal muscle, heart, liver, and brain [25,26,29,30]. Therefore, there is a hormesis response to physical exercise at least partially modulated by oxidative stress levels in several organs including brain [26].

Here we aimed to investigate whether any deleterious cardiovascular and cognitive effects were induced by chronic high-intensity exercise when it does not reach exhaustion. We submitted young adult male mice to forced treadmill exercise for eight weeks at moderate or intense regimens and analyzed physical and cardiovascular adaptive changes as compared to sedentary mice. At the cognitive level, we examined changes in the memory responses and cerebral cortical and hippocampal molecular plasticity markers. Furthermore, we aimed to characterize the brain redox status in both moderate and high-intensity trained mice.

## 2. Materials and Methods

### 2.1. Animals and Experimental Design

Young adult male mice C57BL/6J were used for the study of antioxidant, cognitive and cardiovascular effects after moderate and intense regimens of forced exercise. Forty-five mice were bought to Janvier Labs (Le Genest-Saint-Isle, France) at 6 weeks of age and maintained at the Animal Facility of the University of Barcelona (UB). Animals were housed in Makrolon cages (Techniplast, Buguggiatte, Italy), 5 animals per cage, with free access to food and water in a temperature-controlled room (22 ± 2 °C) with a 12 h light /12 h dark cycle. Following an acclimating period of 2 weeks, they were randomly distributed into three experimental groups: sedentary (SED), moderate physical activity (MOD) and intense physical activity (INT). There was an initial adaptation period of 2 weeks in the treadmill apparatus (LE8710RTS, Panlab, Cornellà, Barcelona, Spain) with progressive intensity in the platform settings. Next, MOD group was forced to run on the platform set at a speed of 15 cm/s and 5 degrees positive slope for 30 min/day, while INT group was forced to run at 30 cm/s and 10 degrees of positive slope for 45 min/day. Training was carried out 5 days/week, Monday through Friday, between 12:00 and 15:00 for 8 weeks. These levels of activity were established in a preliminary study in agreement with previous mouse studies and guidelines [31,32]. The back of the treadmill apparatus included a shock grid that produced a gentle electric shock to induce the animals to keep moving forward. In addition, exercise sessions were observed by a hided experimenter to ensure effective running and to supervise the mice welfare. Once a week, all mice were weighed and examined for any clinical or macroscopic sign of ailment.

Before termination, animals were submitted to behavioral testing and electrocardiographic determinations as described below. At termination, all animals were euthanized with an overdose of inhaled isoflurane anesthetic. Internal organs were examined macroscopically. Heart, aorta, tibia and brain were dissected and processed for analysis as described below. The experimental design and the animal protocols were approved by the UB Ethics Committee of Animal Experimentation (CEEA-UB, #365/19). All procedures were carried out in accordance with the Directive 214/97 of the Generalitat de Catalunya and the European Union (EU) Directive 2010/63/EU for animal experiments.

### 2.2. Cardiovascular Function

All animals were submitted to cardiac determinations to detect adaptive or detrimental changes related to the forced exercise regimens. Heart and aorta were dissected out at the end of the study, at euthanasia. In addition, blood samples were collected with EDTA tubes and centrifuged (2000× *g*, 10 min at 4 °C). The plasma was collected and aliquoted and kept at −80 °C until analysis.

Electrocardiographic parameters. Electrocardiogram (ECG) data were obtained under light anesthesia with 1.5% inhaled isoflurane, to analyze the presence of cardiac physiological changes induced by the exercise treatment (Power-Lab and LabChart v8.0, AD Instruments, Colorado Springs, CO, USA). Analyses were performed at baseline and 48 h after the last training on the treadmill apparatus. Therefore, initial, and final ECG data were obtained and paired analyzed for each mouse. Parameters analyzed included heart rate, PR interval, QRS interval and corrected QT (QTc) interval. At least two recordings per mice were obtained and only high-quality images were used.

NT-proBNP Plasmatic Levels. N-terminal pro B-type natriuretic peptide (NT-proBNP) levels were quantified from 25 µL of plasma with the Human Cardiovascular Disease (CVD) Magnetic Bead Panel 1 (HCVD1MAG-67K, Milliplex Map Kit, Merck Millipore, Burlington, MA, USA). Samples were processed as indicated in the kit and results were measured with a Luminex 200 platform (ThermoFisher Scientific, Waltham, MA, USA).

Heart Hypertrophy. Immediately after postmortem dissection, the heart was immersed in cold physiologic saline, dried in a gaze and weighed. Tibial length was measured to firstly analyze the effect of forced treadmill running. Next, in the absence of exercise effect, tibial length was used to normalize the cardiac weight to avoid possible bias of normalization by body weight.

Aorta Morphology. A segment of descending aorta was fixed in 4% paraformaldehyde and processed for histology by standard procedures. In brief, samples were dehydrated, included in paraffin, and 10 µm transversal tissue sections were stained with picrosirius red. Measurements of each mouse sample were performed in micrographs obtained from the histological aorta slices, using ImageJ software (ImageJ 1.48v, U.S. National Institutes of Health, Baltimore, MD, USA). A set of parameters were quantified, including the lumen area and tunica media area. The later was calculated as: area = (µ/4) × (De^2^ − Di^2^), where De and Di are the diameter external and internal, respectively.

### 2.3. Behavioral Testing

In the seventh week of exercise training, a subgroup of the animals (*n* = 5–6/group) were submitted to behavioral testing to check for the exercise training effects on sensorimotor skills and spatial and recognition learning and memory. Testing was performed on the morning hours at least 3 h before the daily treadmill running treatment and lasted for 5 consecutive days.

Sensory-Motor Functions. Preservation of visual reflex and posterior legs extension reflex were tested by gently holding the animal by its tail while lowering it toward a black surface. Motor coordination and equilibrium were assessed by the distance covered in a horizontal rod on two consecutive trials of 20 s. The test was repeated twice with a progressive increase in the difficulty of the task by using first a square wooden rod (1.3 cm wide) and then a round metal rod (1 cm diameter) [33].

Novel Object Recognition Test (NORT). The assessment of recognition memory relies on the spontaneous tendency of rodents to spend more time exploring a new object than a familiar one. The test was performed as described elsewhere [34]. Briefly, each mouse was placed in the middle of a black rectangular arena (30 × 40 cm) with high walls for 10 min for three consecutive days of habituation. At the 4th day, the animal was allowed a 10 min exploration of two identical objects (A + A) placed equidistant from each other (acquisition trial) and returned to the home cage. Two h later, the animal was allowed to explore two different objects (A + B) where one of the familial objects was replaced for a new one (retention trial). Tests were videotaped and blindly analyzed to record the exploration time of each object. Exploration was defined as the orientation of the nose to the object at a distance of less than 2 cm. Results were calculated as an index of the time the mouse spent exploring the new object versus to the total exploration time.

Novel Object Location Test (NOLT). This test analyzes the rodent ability to recognize a change in position of an object. A higher time exploring the newly located object than the unchanged one is indicative of spatial memory. The test uses the same apparatus and general procedure than the above described NORT, with some differences to test spatial abilities [34]. Here, each mouse was submitted to a 10 min period habituation in the empty arena. The next day, the animal was allowed to explore two identical objects (A1 + A2) placed equidistant from each other (acquisition trial) for 5 min and returned to the home cage. Two h later, the animal was allowed to explore the same objects but one of them had a changed position (A1 + A3) (retention trial). Results were calculated as an index of the time the mouse spent exploring the relocated object versus the total exploration time.

### 2.4. Brain Tissue Analysis

Brain obtained at necropsy was quickly dissected on a cold plate to yield cerebral cortex and hippocampus samples. Tissue samples were snap frozen in liquid nitrogen and preserved at −80 °C until analysis. Prior to analysis, each frozen cerebral cortex sample was powdered under liquid nitrogen and homogenously aliquoted for the different extraction procedures. Hippocampus tissue yield is lower than that of the cortex and was used to confirm the results of neuroplasticity protein levels.

Proteasome Enzymatic Activity. Activation of brain proteasomal enzymatic activity in response to physical exercise was determined in cerebral cortex. Tissue lysates were obtained and further processed as previously described [35]. Proteasomal activity was determined using the Proteasome-Glo™ Assay Systems (Promega, Madison, WI, USA) for chymotrypsin-like activity, trypsin-like activity, and caspase-like activity. Luminescence was measured in an Orion II Microplate Luminometer (Titertek-Berthold, Pforzheim, Germany).

Western Blot Analysis. The levels of specific proteins in cerebral cortex and hippocampus samples were analyzed by Western blot using standard procedures with some modifications as detailed elsewhere [34]. Briefly, 20 µg of denatured protein extracts were electrophoresed in polyacrylamide gels, transferred to PVDF membranes and incubated with specific antibodies for immunodetection of selected proteins. Primary antibodies used were: synaptophysin (DAKO, Agilent, Santa Clara, CA, USA, #A0010), postsynaptic density protein 95 (PSD95) (Millipore, Merck, Darmstadt, Germany, #MAB1598), heat shock protein 70 (HSP70) (Calbiochem, Merck, #HSP01-100UG), proteasome 19S ATPase subunit Rpt6 (Enzo, Farmingdale, NY, USA, #BML-PW9265), proteasome 20S core subunits (Enzo, #BML-PW8155), proteasome 20S subunit β2 (Enzo, #BML-PW9300), 4-hydroxynonenal (4-HNE) (Novus Biologicals, Centennial, CO, USA, #NB100-63093) and nitrotyrosine (Abcam, #ab7048). Secondary antibodies were peroxidase-conjugated (GE Healthcare, Pittsburgh, PA, USA, #NA931 and #NA934). Proteins were visualized using enhanced chemiluminescence detection in a Chemidoc™ Imaging System (Bio-Rad, Hercules, CA, USA). Densitometry value for each sample was normalized to its corresponding value for actin (20–33) (Sigma-Aldrich, Darmstadt, Germany, #A5060) or ß-tubulin (Sigma-Aldrich, #T4026). The semi-quantitative fold differences were identified using Image Lab software v3.0.1 (Bio-Rad). All membranes contained samples from all the experimental groups. Protein level for each sample was calculated relative to the mean of the values of the sedentary control group samples in each membrane.

Quantitative PCR Analysis. Gene expression was determined by real-time quantitative PCR (qPCR) as previously described [34]. In brief, total RNA was obtained from cerebral cortex samples using the mirVana^TM^ miRNA Isolation Kit (Applied Biosystems, Foster City, CA, USA). Random-primed cDNA synthesis was performed using the high-capacity cDNA Reverse Transcription Kit (Life Technologies, Thermo Fisher Scientific, Waltham, MA, USA, #4368814). Gene expression was determined in duplicate samples using TaqMan Fluorescein amidite (FAM)-labeled specific probes, listed in Appendix A, and Quantimix Easy Probe kit (Biotools, Madrid, Spain, #10.601-4149) in an RFX96TM real-time system (Bio-Rad). Data were normalized to actin gene expression using the Comparative Cycle Threshold method (ΔΔCT).

### 2.5. Statistics

Results are expressed as mean ± SEM per group, except the electrocardiographic analysis data that are shown by mouse. Normality of the data was tested with the Shapiro–Wilk test. Body weight and body weight increase curves, and paired electrocardiographic data were analyzed by two-way repeated measures ANOVA (main factors: exercise and time, and their interaction). Validity of using this test was confirmed by small *p*-value of subjects matching where there was significance of factors. All the other data were analyzed for the effect of factor exercise by one-way ANOVA or Kruskal–Wallis test for parametric or non-parametric data, where appropriate. Significant main factors or their interaction were further analyzed with pairwise Bonferroni’s or Dunn’s multiple comparison post hoc test for parametric or non-parametric data, respectively. Statistical outliers were identified with Grubbs’ test (α = 0.05) and removed from the analysis. *p*-values ≤ 0.05 were considered significant. Those for ANOVA factors are indicated into the graphs. Statistical analyses were performed using GraphPad Prism v6.01 (GraphPad Software, San Diego, CA, USA) and IBM SPSS Statistics v23 (IBM Corp., Armonk, NY, USA).

## 3. Results

### 3.1. Mice Exercise Training and Body Weight Evolution

All mice acquired suitable running practice during the 2-week adaptive period and were initially included in the study. However, three mice of the INT group were not able to maintain the intensity demand of the running conditions up to completion and were excluded from the study. The remaining animals did not display any sign of distress or illness. Therefore, the final number in each group was: SED *n* = 15, MOD *n* = 15 and INT *n* = 12.

Body weight and body weight relative to the initial weight curves and ANOVA analysis are displayed in Figure 1a,b, respectively. All groups showed a progressive increase in body weight during at least the first 6 weeks of exercise treatment. Animals in SED group stabilized their body weight at the 7th week. Animals into running regimens showed a decrease in body weight after the 6th week and 7th week for INT and MOD groups, respectively. The at random allocation of mice into the three groups resulted in a slightly different initial weight that, most probably, led to significant differences between SED and MOD body weight curves along second to seventh week (Bonferroni test *p* < 0.05). However, in the relative body weight curve (Figure 1b) all mice displayed a similar body weight increase along the first weeks. Here a difference between SED and INT groups was significant at the last week (Bonferroni test *p* < 0.01). As a whole, exercise treatment decreased body weight progression and this effect was more evident in INT group than MOD group.

### 3.2. Electrocardiography Changes Induced by Exercise

Paired data for each mouse showed some differences between initial and final electrocardiograms. Results for heart rate, PR, QRS and QTc intervals are shown in Figure 2a–d, respectively. Exercise training induced a heart rate decrease in INT group. Accordingly, increased RR interval is shown in paired recordings of a representative mouse. A general differential effect in the PR interval and the QRS interval duration did not reach significance for any of the exercised groups. QTc showed high variability and absence of changes in the paired analysis.

### 3.3. Heart Hypertrophy Induced by Exercise

Heart weights as direct data and data normalized by tibial length are shown in Figure 3a,b, respectively. Tibial length was not modified by physical exercise treatments, as shown in Figure 3c. Therefore, it was a more consistent factor to normalize heart weight than body weight. Heart weight increased with the exercise training according to one-way ANOVA, although there were no differences between group means by post hoc analysis. Significance was reduced to borderline in the calculated ratio with tibial length (ANOVA, *p* = 0.0611). Cardiac hypertrophy was mild in both exercise groups, with heart weight increases of 13.5% and 14.9% in MOD and INT mice, respectively. To further assess putative cardiac changes, NT-proBNP circulating levels were quantified but no differences were detected between groups (Figure 3d). 

### 3.4. Absence of Gross Aortic Changes

The analysis of the aorta morphology performed with tissue slides microscopical images did not detect any effect of exercise training. Measures of the lumen and tunica media thickness are shown as example of absence of gross aortic changes in Figure 4a,b, respectively. Representative images of the descending aorta sections for SED and INT groups used for measurement after picrosirius red staining are shown in Figure 4c. No statistical effect of the exercise regimens was obtained. However, a generally higher variability was shown into the INT animal group.

### 3.5. Exercise Improved Motor Coordination and Did Not Interfere with Cognitive Abilities

All animals tested positive for visual reflex and posterior legs extension reflex, as expected (SED 6/6, MOD 5/5 and INT 6/6). Results of motor coordination are depicted in Figure 5a,b. Exercise training increased motor coordination as shown by higher distance traveled on the squared wooden bar, but the effect was not significant when tested in the more demanding setting of traveling on the metal rod bar. Cognitive responses are depicted in Figure 5c,d. No impairment, but rather a trend to better performances in exercised mice, in both recognition memory and spatial memory was detected withNORT and NOLT assays, respectively. The memory improvement did not reach statistical differences, most probably due to data dispersion. However, the average discrimination indexes of INT group were 216% and 165% of those of SED group for NORT and NOLT, respectively.

### 3.6. Absence of Changes in Markers of Neuronal Function

Protein levels of the presynaptic marker synaptophysin and the postsynaptic marker PSD95 were not modified by MOD or INT regimens of exercise in comparison to SED group. The analysis was performed in cerebral cortex and hippocampus tissues, the main cerebral areas involved in cognitive functions. Results are shown in Figure 6a–f. In addition, we tested the gene expression of the neurotrophic factors brain-derived neurotrophic factor (BDNF), glial cell line-derived neurotrophic factor (GDNF) and vascular endothelial growth factor (VEGF), and the signal transducers cAMP responsive element binding protein 1 (CREB1) and sirtuin 1 (SIRT1) in cerebral cortex. None of them were significantly changed by the exercise training either (Appendix A). However, *Creb1* and *Sirt1* mRNA levels showed a tendency to increase in the average values of the INT group in comparison to the SED values.

### 3.7. Proteasome Activation by Exercise

Catalytic enzymatic activity of the proteasome and mRNA and protein analysis for selected markers of the ubiquitin-proteasome complex were determined in cerebral cortical tissue. Trypsin-like enzymatic activity was increased by physical exercise, while chymotrypsin-like and caspase-like activities remained unchanged. Results are shown in Figure 7a–c.

The gene expression analysis of the 20S catalytic constitutive proteasome and immunoproteasome subunits specific for each enzymatic activity did not show significant changes, nor did the expression of the ubiquitin gene, but there was a consistent trend to increase in several genes after exercise (Appendix A). In brief, mRNA levels of *Psmb6* and *Psmb5* codifying for subunits β1 and β5, associated with caspase-like and chymotrypsin-like activities, respectively, and *Psmb9*, *Psmb10,* and *Psmb8* codifying for the corresponding subunits of the immunoproteasome β1i, β2i and β5i associated with respective caspase-like, trypsin-like, and chymotrypsin-like activities of the INT group increased between 36% and 62% compared to the SED mice group. However, the *Ubc* gene and *Psmb7* gene that codifies for the subunit β2 associated with trypsin-like activity showed similar expression levels in all the experimental groups.

We then analyzed the protein levels of HSP70 a master regulator of ubiquitin-proteasome system, the 19S regulatory particles that serve as proteasome core activator, the 20S catalytic core of the proteasome, and the specific subunit β2 associated with trypsin-like activity. Results are shown in Figure 8a–f. We found a significant decrease in the subunit β2 protein levels in INT mice. Therefore, suggesting that there was a posttranslational response to the increased trypsin-like catalytic activity in INT mice. A trend to decrease in MOD group did not reach significance. The 20S proteasome levels also showed a non-significant trend to decrease in INT group and no changes were detected in HSP70 and 19S proteasome.

### 3.8. Maintenance of Redox Homeostasis under Exercise Training

Levels of the established oxidative stress markers 4-HNE and nitrotyrosine were determined in protein extracts of cerebral cortical tissue. The results shown in Figure 9a,b demonstrated the absence of increased peroxidated and nitrosylated proteins in MOD and INT groups that would be detected by increased 4-HNE- and nitrotyrosine-labeled proteins, respectively. We also analyzed the mRNA levels of the transduction factor nuclear factor erythroid 2-related factor 2 (Nrf2) and two first-line antioxidant enzymes, Mn superoxide dismutase (SOD2) and catalase. Results are shown in Figure 9c–e. *Nfe2l2* and *Sod2* expression was maintained, but C*at* expression was significantly enhanced in INT mice.

## 4. Discussion

Young adult male mice were used in this study to unveil if any deleterious effect was triggered by intense physical exercise in healthy animals. According to an approximate age equivalence between humans and mice [36], the 10-week-old animals at week 0 of exercise training correspond to 20-year-old humans, while 18-week-old mice at termination corresponds to 35-year-old humans. Therefore, we are dealing with chronic exercise treatment that extends over the human equivalent of 15 years in young adulthood. We found that mice submitted to training showed dose-related effects indicative of fitness, such as slightly decreased body weight and improved motor coordination in intense but not moderate exercise. In a similar design, eight weeks of treadmill exercise in young mice of the same strain were reported to induce a reduction in body weight, fat mass, and adipocyte size while increasing grip strength and skeletal muscle mitochondria biogenesis markers [37]. Of note, our intensive training level was below strenuous conditions because even though three of 15 mice could not keep up with the protocol pace, no general behavioral alterations were observed in the mice remaining in the study.

Cardiovascular changes were more prominent in the group submitted to intense training. These mice showed a resting heart rate significantly lower than the rate before exercise, whereas those of sedentary mice and mice submitted to moderate training were unchanged after the 8-week study period. A decrease in resting heart rate has been reported after chronic exercise in mice, both wild-type and disease mouse models [38,39]. In the absence of pathological ECG changes, a decrease in resting heart rate is generally considered an adaptive change promoted by physical endurance that is documented at all ages [40,41] and is generally associated with health-related physical fitness [42].

Increased heart weight was another relevant change induced by treadmill exercise in both moderate and intensive trained mice. Cardiac hypertrophy was mild in either mice group. This increase was in accordance with the ventricular hypertrophy previously reported for chronic uphill treadmill running in the C57CL/6J male mice [43]. These authors found a significant left ventricle wall thickness increase that accounted for 12–17% of the ventricular weight increase. Furthermore, intensity-controlled treadmill running is considered a characterized model of physiological cardiac hypertrophy, and it may be used to study adaptive human responses to exercise training [44]. Physiological cardiac hypertrophy has been reported in trained athletes of all ages [45,46]. These adaptive changes are generally considered safe and recent views propose maintaining exercise training in individuals with pathological cardiomyopathy [47].

To confirm the physiological nature of exercise-induced cardiac remodeling, we quantified NT-proBNP plasma levels at the end of the two-months training protocol. NT-proBNP is the gold standard biomarker for hemodynamic overload, and its concentration is heightened in conditions such as heart failure [48,49]. Even mild B-type natriuretic peptide elevations (i.e., BNP and NT-proBNP) predicted the risk of cardiovascular events in the Framingham Offspring Study [50]. Similar circulating NT-proBNP levels in the plasma of sedentary and exercised mice indicate physiological exercise-induced hemodynamic cardiac changes.

No changes in descending thoracic aorta thickness were detected in treadmill-trained mice in comparison to sedentary animals. Data from athletes suggest that arterial remodeling takes place with a larger lumen and decreased wall thickness after long-term endurance exercise. Nevertheless, such arterial remodeling has been reported mainly in carotid, femoral and brachial arteries. Data analyzing aortic changes in human athletes is scarce and less consistent, so there is no current consensus in the field [51]. Altogether, we can speculate that mice underwent a physiological cardiovascular adaptation in accordance with experimental studies that report the beneficial effects of moderate exercise in some cardiomyopathies and great vessel pathologies [52,53]. In contrast, a rat model of long-term strenuous exercise has recently been shown to promote deleterious vascular remodeling [54].

The preserved recognition memory and spatial memory and the maintained levels of the synaptic markers synaptophysin and PSD95 allowed discarding harmful brain effects of the intense treadmill training. Moreover, the lack of significant improvement in memory responses after either the moderate or the intense protocol was not surprising in young animals. BDNF is a neurotrophic factor demonstrated to mediate the neuroprotective effects of physical exercise [8]. Its expression was not increased in the brain, but we performed all analyses with tissues collected 48 h after the last training session. Similarly, we can speculate that peripheral levels of BDNF that are mainly generated by skeletal muscle would be maintained or rather decreased in the resting state, as reported in young men after chronic exercise [55]. The cognitive benefit of physical activity has been indisputably recognized in old age; it is known that a physically active lifestyle contributes to optimizing brain function and avoiding age-related cognitive decline [5]. However, cognitive benefits in regular sport-practicing young adults have been poorly analyzed in comparison to children and older adults [56]. Less obvious benefices are expected in those in their 20s because the brain is at peak performance age [57]. Nevertheless, modest effects have been demonstrated following aerobic training interventions in young adults on attention, processing speed, executive function, and less consistent effects on memory [58,59].

Redox homeostasis was also preserved in the brain of exercised mice, as demonstrated by regular levels of the two biomarkers of protein oxidation, 4-NHE, and nitrotyrosine. Oxidative modification of proteins has been reported as the primary effect of oxidative stress mediated by strenuous physical exercise [60]. The absence of oxidative damage in the highly vulnerable nerve tissue [61] suggests a general systemic adaptation to oxidative challenge [62] in the exercised mice.

Interestingly, we found an increase in proteasome activity and catalase expression, two first-line mechanisms of enzymatic defense against oxidative stress and cell damage. The intracellular protein complexes 26S proteasomes, named according to the sedimentation coefficient, are the main ones responsible for the degradation of unneeded or damaged proteins. Interventions of endurance physical exercise in healthy young men have triggered parallel increases in 26S proteasome peptidase activity and oxidized protein degradation in the skeletal muscle [63]. We analyzed the proteasome components 19S and 20S. The lack of changes in protein levels of the regulatory cap component 19S proteasome and in the proteolytic signaling hub HSP70 [64] suggests that the effects were at the level of the catalytic 20S proteasome or immunoproteasome. We found an increase in trypsin-like enzymatic activity in the cerebral cortex, whereas other authors have reported also increases in chymotrypsin-like and caspase-like enzymatic activities in the hippocampus of mice submitted to voluntary wheel running [65]. The proteasome is a very dynamic system; hence, the trypsin-like enzymatic activity was similarly increased in both exercise groups. We can speculate that such activity could have been higher in the intensely trained group if the protein levels of the corresponding β5 subunit were not decreased as a feedback adjustment. The catalase gene expression also showed a higher increase in the intensive mice in comparison to the moderate ones, which was significantly higher in the former group compared to the sedentary group. Catalase is a very potent antioxidant enzyme that degrades hydrogen peroxide, and its enzymatic level is mainly determined by gene expression modulation [66]. In this regard, the practice of leisure sport has shown to increase catalase gene expression in the peripheral whole blood of middle-aged men, in parallel with memory improvement and plasma oxidative markers decrease [55,67]. Furthermore, the absence of changes in the mitochondrial SOD2 gene expression in the exercise mice brain would indicate the maintenance of redox homeostasis in this organelle. Similarly, other authors have reported an increase in catalase enzymatic activity but not SOD activity in the hippocampus of mice submitted to wheel running [68]. Finally, we did not detect expression changes of the transduction factor Nrf2 after moderate or intense training. Nr2f regulates antioxidant defense, including the basal and inducible levels of catalase and proteasome [69]. Most probably, the remaining oxidative stimulus was too low to maintain higher Nf2r gene expression even though it could maintain the nuclear translocation of the Nr2f protein.

The maintenance of proteasome and catalase activation in the absence of oxidative stress suggests the establishment of resilience mechanisms against present and future brain damage. In addition to contributing to the oxidized proteins recycling, the proteasome also hydrolyzes the aberrant proteins that would lead to neurodegenerative diseases such as amyloid-β in Alzheimer’s disease [70]. Active catalase will also contribute to decreasing age-related oxidative stress that underlies inflammaging and increases neurodegenerative risks [10]. Furthermore, these advantages in brain health are paralleled with physiological adaptive cardiac changes to chronic aerobic exercise in this athlete mice model. Improved peripheral organ fitness by exercise may also enhance resilience against brain damage and Alzheimer’s disease [71].

This study has limitations, as we addressed the effects of relatively low and high intensities of exercise, but the consequences of strenuous endurance training, which sustains hemodynamic overload, remain unexplored. In addition, it has been performed in male mice but not in females. However, modulation of the oxidative stress responses to physical exercise seems similar in men and women [72]. Moreover, no differences in cardiac risk in life-long masters’ athletes have been reported, despite sex differences in cardiac morphology and physiology [73]. However, literature about exercise effects on women or female animals is scarce and more studies are warranted, which are beyond the scope of this project.

## 5. Conclusions

In summary, we have shown that young, healthy male mice adapt to two different intensity exercise regimes, achieving better motor coordination skills with the high-intensity protocol. Both exercise intensities triggered cardiac adaptation, with no hemodynamic overload but with bradycardia in the highly exercised mice. Memory responses were preserved, and no exercise-induced changes of the synaptic markers synaptophysin and PSD95 were detected in brain samples, implying no brain circuitry alterations. In cerebral cortical tissue, increased proteasome activity and catalase antioxidant levels, together with the lack of oxidative stress damage, suggest an improvement in brain health and resilience capacity, particularly in the high-intensity exercised mice.

## Figures and Tables

**Figure 1 antioxidants-11-01891-f001:**
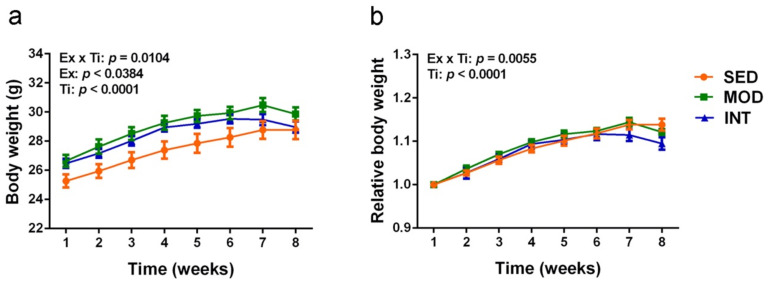
Daily treadmill training induced a body weight decrease in mice. (**a**) Body weight curves; (**b**) relative body weight curves. Experimental groups: SED, sedentary; MOD, moderate training; INT, high-intensity training. Values are mean ± SEM ((**a**,**b**) SED *n* = 15, MOD *n* = 15, INT *n* = 12). ANOVA statistics: Ex, factor exercise; Ti, factor time; see text for post hoc results.

**Figure 2 antioxidants-11-01891-f002:**
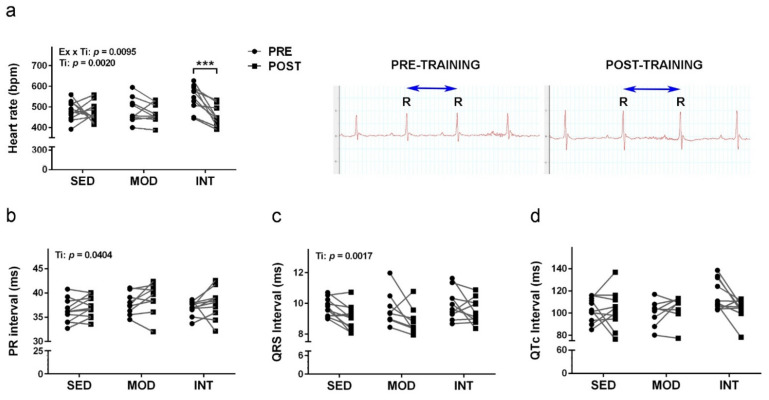
Exercise training induced a resting heart rate decrease. Individually paired data of electrocardiograms previous (PRE) and posterior (POST) to the training period. (**a**) Heart rate and paired electrocardiograms of a representative INT mouse with indication of RR interval; (**b**) PR interval; (**c**) QRS interval; (**d**) QTc interval. Experimental groups: SED, sedentary; MOD, moderate training; INT, high-intensity training. Values are individual mouse data for each group ((**a**,**b**) SED *n* = 11, MOD *n* = 9, INT *n* = 10; (**c**,**d**) SED *n* = 11, MOD *n* = 8, INT *n* = 10). ANOVA statistics: Ex, factor exercise; Ti, factor time. Bonferroni test *** *p* < 0.001.

**Figure 3 antioxidants-11-01891-f003:**
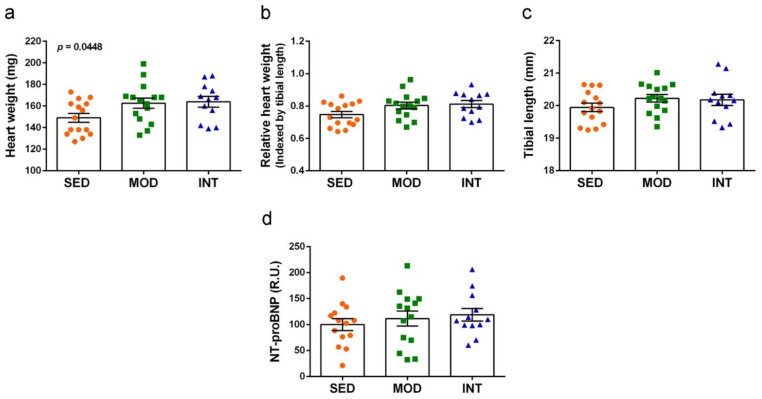
Exercise training induced mild cardiac hypertrophy. (**a**) Heart weight; (**b**) relative heart weight; (**c**) tibial length; (**d**) *N*-terminal pro B-type natriuretic peptide (NT-proBNP). Experimental groups: SED (orange circles), sedentary; MOD (green squares), moderate training; INT (blue triangles), high-intensity training. Values are mean ± SEM ((**a**–**c**) SED *n* = 15, MOD *n* = 15, INT *n* = 12); (**d**) SED *n* = 14, MOD *n* = 14, INT *n* = 12). Statistics: There was a significant ANOVA effect of exercise treatment without post hoc test significances in (**a**) and a borderline significant effect in (**b**).

**Figure 4 antioxidants-11-01891-f004:**
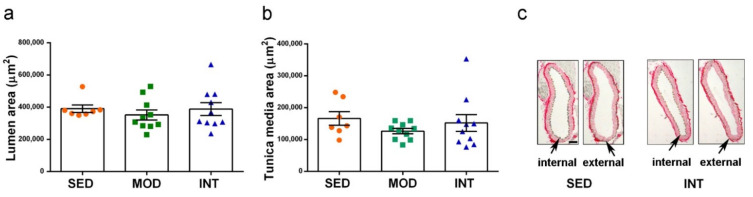
Exercise training did not induce gross morphological changes in the aorta. (**a**) Area of the aorta lumen; (**b**) area of a transversal section of the tunica media; (**c**) representative histological images of the aorta section with arrows pointing to dots indicating the internal and external limit of the tunica media. Experimental groups: SED (orange circles), sedentary; MOD (green squares), moderate training; INT (blue triangles), high-intensity training. Values are mean ± SEM ((**a**,**b**) SED *n* = 7, MOD *n* = 10, INT *n* = 10). Scale bar = 100 µm in (**c**).

**Figure 5 antioxidants-11-01891-f005:**
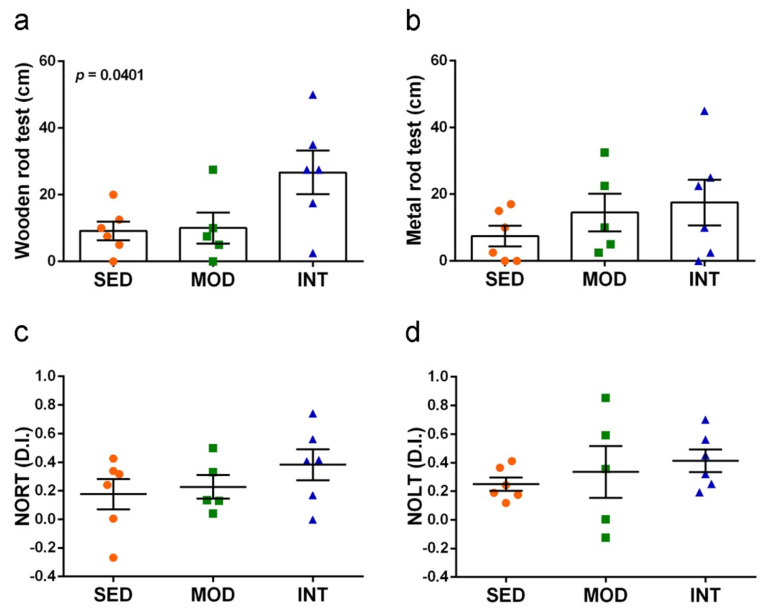
Exercise training improved motor coordination and maintained learning and memory skills. (**a**) Distance traveled on a squared wooden rod; (**b**) distance traveled on a round metal rod; (**c**) discrimination index (D.I.) in the novel object recognition test (NORT); (**d**) discrimination index (D.I.) in the novel object location test (NOLT). Experimental groups: SED (orange circles), sedentary; MOD (green squares), moderate training; INT (blue triangles), high-intensity training. Values are mean ± SEM ((**a**–**d**) SED *n* = 6, MOD *n* = 5, INT *n* = 6). Statistics: There was a significant ANOVA effect of exercise treatment in (**a**) without post hoc test significance.

**Figure 6 antioxidants-11-01891-f006:**
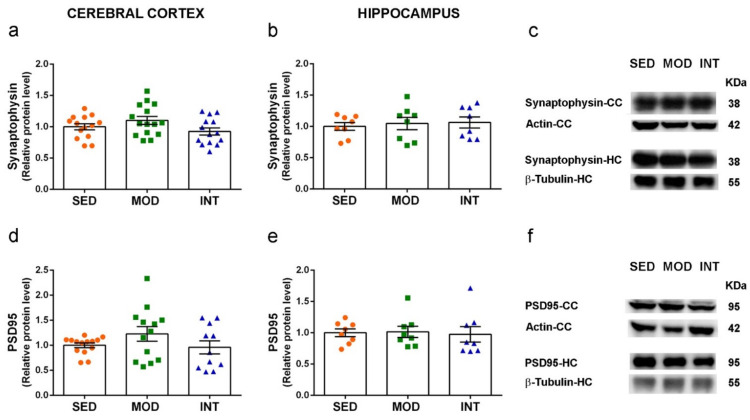
Synaptic markers synaptophysin and PSD95 were not modified by exercise in young mice. (**a**–**c**) Levels of synaptophysin in the cerebral cortex (CC) and hippocampus (HC), as indicated, and representative blots of experimental groups; (**d**–**f**) levels of PSD95 in cerebral cortex and hippocampus and representative blots. Experimental groups: SED (orange circles), sedentary; MOD (green squares), moderate training; INT (blue triangles), high-intensity training. Values are mean ± SEM ((**a**) SED *n* = 14, MOD *n* = 15, INT *n* = 14; (**b**,**e**) SED *n* = 8, MOD *n* = 8, INT *n* = 8; (**d**) SED *n* = 14, MOD *n* = 13, INT *n* = 11).

**Figure 7 antioxidants-11-01891-f007:**
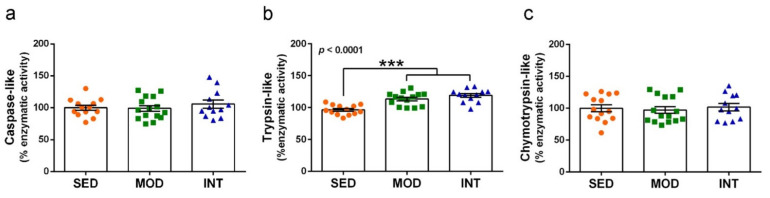
Proteolytic activity of the proteasome was increased by exercise. Caspase-like enzymatic activity (**a**), trypsin-like activity (**b**), and chymotrypsin-like activity (**c**) were determined in cerebral cortical tissue. Experimental groups: SED (orange circles), sedentary; MOD (green squares), moderate training; INT (blue triangles), high-intensity training. Values are mean ± SEM ((**a**) SED *n* = 13, MOD *n* = 15, INT *n* = 12; (**b**) SED *n* = 13, MOD *n* = 14, INT *n* = 12; (**c**) SED *n* = 14, MOD *n* = 15, INT *n* = 12). Statistics: There was a significant ANOVA effect of exercise treatment in (**b**) with post hoc test *** *p* < 0.001 for both MOD and INT in comparison to the SED group.

**Figure 8 antioxidants-11-01891-f008:**
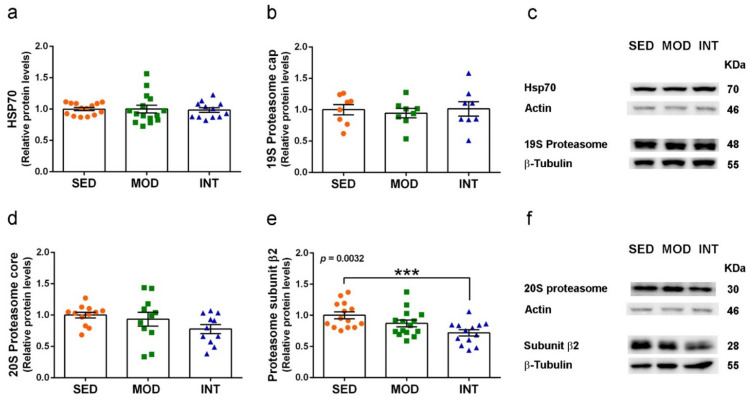
Protein levels of proteasome subunits were modified by exercise. HSP70 (**a**), 19S proteasome cap (**b**), and representative blots of each experimental group (**c**); 20S proteasome core (**d**) proteasome subunit β2 (**e**) and representative blots (**f**). Proteins were determined in cerebral cortical tissue. Experimental groups: SED (orange circles), sedentary; MOD (green squares), moderate training; INT (blue triangles), high-intensity training. Values are mean ± SEM ((**a**) SED *n* = 14, MOD *n* = 15, INT *n* = 12; (**b**) SED *n* = 8, MOD *n* = 8, INT *n* = 8; (**d**) SED *n* = 12, MOD *n* = 11, INT *n* = 11; (**e**) SED *n* = 14, MOD *n* = 15, INT *n* = 13). Statistics: There was a significant ANOVA effect of exercise treatment in (**f**) with post hoc test *** *p* < 0.001 in comparison to the SED group.

**Figure 9 antioxidants-11-01891-f009:**
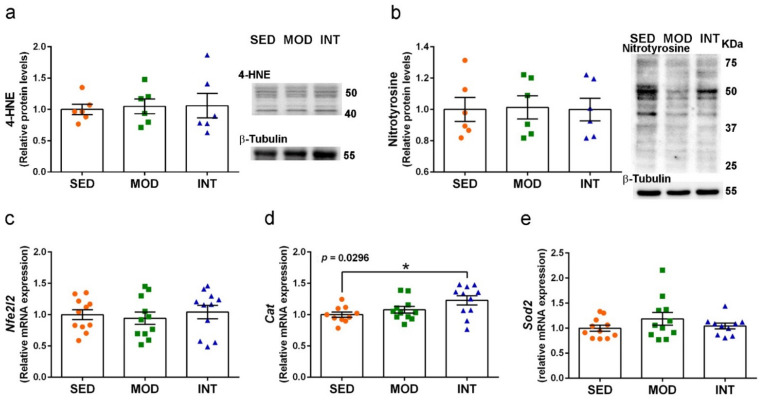
Increase in antioxidant defense and absence of oxidative stress in exercised mice. Levels of proteins labeled with 4-HNE (**a**) and nitrotyrosine (**b**), including corresponding representative blots; mRNA levels of *Nfe2l2* (**c**), *Cat* (**d**), and *Sod2* genes (**e**). Analyses were performed on cerebral cortical tissue. Experimental groups: SED (orange circles), sedentary; MOD (green squares), moderate training; INT (blue triangles), high-intensity training. Values are mean ± SEM ((**a**,**b**) SED *n* = 6, MOD *n* = 6, INT *n* = 6; (**c**,**e**) SED *n* = 11, MOD *n* = 11, INT *n* = 11; (**d**) SED *n* = 11, MOD *n* = 11, INT *n* = 10). Statistics: There was a significant ANOVA effect of exercise treatment in (**d**) with post hoc test * *p* < 0.05 in comparison to the SED group.

## Data Availability

All data are included in the article and Appendix A.

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
