# Peer review of "Antioxidant Molecular Brain Changes Parallel Adaptive Cardiovascular Response to Forced Running in Mice"

_antioxidants, 2022, doi:10.3390/antiox11101891_

Round 1

Reviewer 1 Report

The current MS presents interesting results related to the effect of moderate and high-intensity physical activity on the Brain and heart changes. The results despite seeming still speculative, especially regarding cardiovascular adaptation, will be very useful to guide several other further studies, that aim to better explain adaptative mechanisms related to protective or harmful effects. 

I would suggest that the authors show some results related to body composition, before/after the physical activity regimens. Considering that the body weight was similar, I was wondering if correlating some variabilities that occurred in other body compounds will be useful to explain the heart hypertrophy.

Also, considering that heart hypertrophy seems the most consistent change observed in the cardiovascular system, exploring this variability would be interesting. Thus, as shown in neuronal function, molecular biomarkers related to heart function are needed to suggest that the slight heart hypertrophy observed is not dangerous. Markers related to apoptosis/fibrosis also would be interesting.

Author Response

The current MS presents interesting results related to the effect of moderate and high-intensity physical activity on the Brain and heart changes. The results despite seeming still speculative, especially regarding cardiovascular adaptation, will be very useful to guide several other further studies, that aim to better explain adaptative mechanisms related to protective or harmful effects. 

Point 1

I would suggest that the authors show some results related to body composition, before/after the physical activity regimens. Considering that the body weight was similar, I was wondering if correlating some variabilities that occurred in other body compounds will be useful to explain the heart hypertrophy.

Response: The analysis of other body compounds would be interesting in order to better understand the overall effect of exercise in mice physiology and adaptation to exercise, as it is suggested by the Reviewer. Nevertheless, we did not find any differences in tibia length due to exercise and further analysis in skeletal muscle cannot be performed because we did not collect these samples at the euthanasia time. We discussed the heart hypertrophy as a physiological remodeling as is described in the literature as the “athlete’s heart”. See references #45 (Pieles et al., 2020, DOI: 10.1002/clc.23417) and #46 (Kusy, K. et al., 2021, 10.1016/j.echo.2021.06.009).  

Point 2

Also, considering that heart hypertrophy seems the most consistent change observed in the cardiovascular system, exploring this variability would be interesting. Thus, as shown in neuronal function, molecular biomarkers related to heart function are needed to suggest that the slight heart hypertrophy observed is not dangerous. Markers related to apoptosis/fibrosis also would be interesting.

Response: To address the reviewer comment we have analyzed NT-proBNP in plasma, the gold standards for cardiac hemodynamic overload and heart failure. We have included these results in figures 3d. No changes in blood concentration were detected after the exercise protocols in NT-proBNP indicating that that no adverse cardiac remodeling was present. We have included a paragraph in the Discussion (lines 470-475) and a phrase in Conclusions (lines 544-545). Also, we have included a new sentence in the limitations section to emphasize that we did not reach the levels of strenuous endurance training that trigger hemodynamic overload (lines 532-534).

Because we did not find changes in NT-proBNP, a specific cardiac biomarker, we did not further pursue putative changes in apoptosis or fibrosis markers due to their lack of specificity.

Reviewer 2 Report

The authors aimed to analyzed deleterious cardiovascular and cognitive  effects  induced by chronic high intensity exercise when it does not reach exhaustion in young adult male mice submitted to exercise for eight weeks at moderate or high intensity regimens compared to sedentary mice. 

The paper is intersting,  clearly written however some minor concerns are due:

1.why the authors don't use the MWM in order to test learning and memory in the exercised mice? 

2. The author's have loocked at the expression of circulating BDNF, whose expression is associated to better neuroplasticity and reduced neurodegenerative disease in older trained subjects?;  could the authors discuss this point?

3. the authors claimed that intense exercise affects proteasome pathway in brain, that in turn is involved in longevity. They evaluated  VEGF expression in brain? May be interesting also to know the AMPK, Citrate Syntase and PGC1a protein expression in muscle of trained mice compared to untrained, also in association with weight reduction due to the fact that aerobic exercise ativated the expression of these proteins in skeletal muscle, and the  expression in turn is associated to better cardiorespiratory fitness; 

4. the authors loocked at the plasma oxidative  status in trained compared to untrained mice? may be interested to loock at T0 and after 4 and 8-w of training in that exercise adaptation could influence the results after 8-w of training; please discuss this point.

Author Response

Manuscript ID: antioxidants-1878152

Response to Reviewer 2 Comments

The authors aimed to analyzed deleterious cardiovascular and cognitive effects induced by chronic high intensity exercise when it does not reach exhaustion in young adult male mice submitted to exercise for eight weeks at moderate or high intensity regimens compared to sedentary mice.

The paper is intersting, clearly written however some minor concerns are due:

  1. why the authors don't use the MWM in order to test learning and memory in the exercised mice?

Response: We agree that MWM is an excellent test for spatial learning and memory in mice and we use it in our laboratory of animal behavior. However, here the animals were trained in a physical task that might interfere with swimming performance and capacity to discern the landmarks while swimming. In our experience, the combined use of NORT and NOLT allows also detecting any learning and memory deficiency in both recognition memory and spatial memory. In addition these tests are less stressful for the animals than MWM.

  1. The author's have loocked at the expression of circulating BDNF, whose expression is associated to better neuroplasticity and reduced neurodegenerative disease in older trained subjects?; could the authors discuss this point?

Response: We had analyzed the levels of BDNF RNA in the cerebral cortex to check for the neurotrophic brain status and there were no significant changes with exercise training. However, peripheral BDNF is mainly generated by skeletal muscle in response to acute exercise and its levels return to baseline within a short time. Termination samples were obtained 48 h after the last training. Therefore we did not expect a peripheral BDNF increase. However we agree the relevance of BDNF and other trophic factors in the neuroprotective effect of chronic exercise against neurodegenerative disease. We added a comment in Discussion (lines 447-454).

  1. the authors claimed that intense exercise affects proteasome pathway in brain, that in turn is involved in longevity. They evaluated VEGF expression in brain? May be interesting also to know the AMPK, Citrate Syntase and PGC1a protein expression in muscle of trained mice compared to untrained, also in association with weight reduction due to the fact that aerobic exercise ativated the expression of these proteins in skeletal muscle, and the expression in turn is associated to better cardiorespiratory fitness;

Response: In order to answer the Reviewer, we evaluated the Vegfa RNA levels in cerebral cortex. Results are displayed in the Supplementary Figure 1Sc. One-way ANOVA showed absence of significant effects induced by exercise treatment [F (2,30) =1.793, P = 0.1838]. Absence of changes in VEGF expression was indicated in Section 3.6. Absence of changes in markers of neuronal function. Proteasome activation is considered a relevant mediator of intracellular benefices against neurodegenerative diseases. It is also activated in the skeletal muscle. We agree on the interest of the proposed proteins and others involved in the mitochondrial and metabolic rejuvenation of skeletal muscle cells through chronic physical exercise. However we did not kept gastrocnemius or soleus tissue samples because their study was beyond the scope of the present work.

  1. the authors loocked at the plasma oxidative status in trained compared to untrained mice? may be interested to loock at T0 and after 4 and 8-w of training in that exercise adaptation could influence the results after 8-w of training; please discuss this point.

Response: We performed a preliminary determination of malondialdehyde (MDA) in plasma samples obtained at termination, as we did not preserve plasma samples at t = 0 or t = 4 weeks. We have previously demonstrated that chronic physical exercise drecrease MDA plasma levels in men, although the effect was mild, whereas no change was obtained in carbonylated proteins (#55 De la Rosa et al., 2019, DOI: 10.1038/s41598-019-40040-8). We used the commercial kit MAK086-5 (Sigma-Aldrich) and followed the manufacturer’s intructions. However all mouse samples showed undetectable MDA levels that are not reliable to quantify, but demonstrated absence of oxidative stress difference between the experimental groups. This is in agreement with brain tissue results. Nevertheless, we would prefer not to include these results in the paper, because we are concerned that the kit was not sensitive enough for our samples with very low MDA levels.

Reviewer 3 Report

This is a well-crafted manuscript. In this study the authors demonstrated that young healthy male mice adapt to two different intensity exercise regimes, achieving better motor coordination skills and cardiac adaptation with the high-intensity protocol and no brain circuitry alterations. There are some minor suggestions for the authors.

1.     In Materials and Methods, Animals and Experimental Design: Is the timing of the animal's movement fixed, and does it have any effect on the results?

2.     How many days a week does the animal exercise and does it rest?

3.     In the legends, please add the number of rats in each group.

4.     In Materials and Methods, Brain tissue analysis: why proteasome enzymatic activity and quantitative PCR determined the cerebral cortex, while WB determined the cerebral cortex and hippocampus?

5.     Figure 4c, the lack of rulers on the graph.

6.     Please double-check the details in this manuscript, such as line 265 and line 266, should it be Figure 3abc.

Author Response

Manuscript ID: antioxidants-1878152

Response to Reviewer 3 Comments

This is a well-crafted manuscript. In this study the authors demonstrated that young healthy male mice adapt to two different intensity exercise regimes, achieving better motor coordination skills and cardiac adaptation with the high-intensity protocol and no brain circuitry alterations. There are some minor suggestions for the authors.

  1. In Materials and Methods, Animals and Experimental Design: Is the timing of the animal's movement fixed, and does it have any effect on the results?

Response: Daily training was set at 30 min for MOD group and 45 min for SED group and took place between 12:00 and 15:00. We added this information in the Section 2.1. Animals and Experimental Design. The timing of the exercise intervention was fixed throughout the study for all mice because it may interfere with the rodent’s circadian rhythm and thus influence the level of running. Training in the early afternoon allows the animals to rest in the morning after their nighttime activity period and allows them to continue with their daytime period of low activity after the session.

  1. How many days a week does the animal exercise and does it rest?

Response: The mice exercised 5 days a week from Monday to Friday and rested on Saturday and Sunday. We added this information in the Section 2.1. Animals and Experimental Design.

  1. In the legends, please add the number of rats in each group.

Response: We added N values for each group in all figure legends of the Article and Supplementary Materials.

  1. In Materials and Methods, Brain tissue analysis: why proteasome enzymatic activity and quantitative PCR determined the cerebral cortex, while WB determined the cerebral cortex and hippocampus?

Response: We used the same regional brain tissue in the diverse determinations to draw well-founded conclusions. That is, we used cerebral cortical tissue for the analysis of protein levels or gene expression of specific neuroplasticity markers (Fig 6 and S1), proteasome enzymatic activity (Fig 7), protein levels and/or gene expression of specific components of the ubiquitin-proteasome system (Fig 8 and S2), and protein levels or gene expression of oxidative stress markers and enzymes (Fig 9). Hippocampal tissue is scarce in the mouse and there was not enough tissue to perform all these analyses. These analyses require different extraction procedures with adequate extract volume. Therefore, we powdered the frozen cerebral cortex of each mouse under liquid nitrogen to obtain homogenous aliquots for proteins, RNA and enzymatic biochemistry analysis, respectively. This is a usual procedure (see detailed description in Masuo et al., 2011; DOI: 10.1007/978-94-007-0828-0_11; Section 11.3 Preparation of Powdered Brain/Regions). Both the hippocampus and cerebral cortex are target areas in cognitive loss and neurodegenerative processes. However, circuits in the hippocampus differ from those in the cerebral cortex, and we therefore confirmed the main results involving neuronal markers in both brain regions, cerebral cortex and hippocampus. The whole hippocampus was used for Western blotting. We added a comment in the Section 2.4. Brain tissue analysis.

  1. Figure 4c, the lack of rulers on the graph.

Response: We added the corresponding Scale bar in Figure 4c.

  1. Please double-check the details in this manuscript, such as line 265 and line 266, should it be Figure 3abc.

Response: We revised and corrected mistaken details throughout the manuscript.

Round 2

Reviewer 1 Report

The authors answered all questions and included adequate material/results to improve the quality of the MS.